

# Evaluating *COI* and ITS2 dual barcoding for molecular delimitation and taxonomic insights in *Arenosetella* Wilson, 1932 (Harpacticoida: Ectinosomatidae) along Turkish Coasts

Dilara Bakmaz[1], Serdar Sönmez[2] and Ertan Mahir Korkmaz[1]

[1] Department of Molecular Biology and Genetics, Faculty of Science, Sivas Cumhuriyet University, Sivas, Turkey
[2] Department of Biology, Faculty of Science and Letters, Adıyaman University, Adıyaman, Turkey

Corresponding author
Serdar Sönmez,
sonmezserdar@gmail.com

## ABSTRACT

**Background**. Accurate species delimitation is essential in morphologically conservative taxa such as harpacticoid copepods, in which cryptic diversity may go unnoticed without molecular data. The genus *Arenosetella*, common along the Turkish coastline, comprises two species: *Arenosetella germanica* and *A. lanceorostrata*, with overlapping ranges and subtle morphological differences. This study aimed to assess species boundaries and uncover potential hidden diversity within *Arenosetella* using the dual-marker DNA barcoding approach.

**Methods**. Specimens of *Arenosetella* were collected from the Mediterranean, Aegean, and Black Sea coasts of Türkiye. Nuclear DNA from a total of 46 individuals were amplified and sequenced for both mitochondrial cytochrome oxidase I (*COI*) and nuclear internal transcribed spacer 2 (ITS2) markers. *COI* sequences were analysed for haplotype diversity, phylogenetic relationship, and species delimitations. ITS2 sequences were subjected to evaluation with regard to nucleotide diversity, secondary structure, and compensatory base changes (CBCs), using both sequence- and structure-based approaches. The concatenated dataset and species tree reconstruction (Star-BEAST2) were employed to test gene tree-species tree congruence.

**Results**. The *COI* analyses revealed a high level of haplotype diversity (21 haplotypes) and the presence of three molecular operational taxonomic units (MOTUs) within *A. germanica* and one MOTU within *A. lanceorostrata*, consistent with the geographic distribution patterns. ITS2 sequences exhibited relatively more conservation with nine haplotypes. These sequences revealed informative structural variation, including CBCs among candidate species. The species delimitation approaches reliably supported the identification of four to seven MOTUs, which corresponded to geographic populations. The analyses of the concatenated dataset supported four well-supported candidate species, and yielded congruent species trees, with high posterior probabilities. Morphological comparisons among MOTUs revealed subtle differences in female P5 structure and anal somite ornamentation among *A. germanica* lineages, while *A. lanceorostrata* MOTUs were morphologically indistinguishable.

**Conclusion**. This study provides the first integrative application of *COI* and ITS2 barcoding in *Arenosetella* and within Harpacticoida overall, combining DNA sequences and structure, and morphological data for species delimitation. The results demonstrate

that *COI* is effective for detecting geographic differentiation and haplotype diversity, whereas ITS2 contributes structural resolution and potential markers of reproductive isolation through CBCs. These findings suggest the presence of a species complex within *A. germanica* and confirm the distinct status of *A. lanceorostrata*. Dual-marker barcoding, particularly incorporating ITS2 secondary structure, represents a valuable tool for taxonomic studies in morphologically conservative copepod groups.

## INTRODUCTION

Species are the fundamental units of biodiversity, forming the basis for evolutionary studies and shaping our understanding of ecological interactions and relationships (*Dayrat, 2005*; *De Queiroz, 2007*). Accurate species delimitation is crucial for biodiversity assessments and systematic studies, particularly in groups like Copepoda in which morphological similarities often obscure species boundaries. These small crustaceans are among the most abundant metazoans, inhabiting diverse aquatic environments and playing key roles in ecosystems as primary and secondary consumers (*Huys & Boxshall, 1991*; *Thorp & Rogers, 2011*).

The order Harpacticoida is particularly diverse in meiobenthic communities and includes taxa with considerable ecological and evolutionary importance. Within this group, the genus *Arenosetella* Wilson, 1932 (Harpacticoida: Ectinosomatidae) is composed of small, benthic copepods adapted to interstitial habitats in coastal sediments (*Sönmez, Sak & Karaytuğ, 2016*). *Arenosetella germanica* (*Kunz, 1937*) and *Arenosetella lanceorostrata* (*Sönmez, Sak & Karaytuğ, 2016*) are widely distributed along the Turkish coasts (*Kabaca, 2024*). *A. germanica* has been documented to exhibit intraspecific morphological variation among the populations (*Mielke, 1975*; *Mielke, 1986*) particularly concerning leg setal formulae, which Mielke interpreted as within-species variability, highlighting the need for integrative approaches to clarify its species boundaries. Such approaches are particularly valuable for uncovering cryptic diversity, especially when morphological data alone are insufficient.

Morphologically conservative groups such as harpacticoid copepods may require the use of molecular tools to facilitate the delineation of species boundaries. The mitochondrial cytochrome oxidase subunit I (*COI*) gene is a widely used DNA barcode for species delimitation (*Hebert et al., 2003a*; *Bucklin et al., 2010*). In conjunction with *COI*, the nuclear internal transcribed spacer 2 (ITS2; a non-coding spacer region within the nuclear ribosomal DNA) region provides valuable insights into species- and population-level divergences, attributable due to its high sequence variability among species and conserved structural features within species (*Yao et al., 2010*; *Coleman, 2003*). However, they differ fundamentally in their molecular evolution and selective constraints. The COI subjects to functional constraints and purifying selection, though its high mutation rate in certain

regions makes it valuable for detecting interspecific variation. In contrast, the ITS2 evolves more rapidly due to relaxed selective pressure, with concerted evolution homogenizing its sequence within species (*Dover, 1982*). These differences influence their utility in resolving taxonomic boundaries and phylogenetic relationships, with COI often preferred for deep divergences and ITS2 for closely related species. Furthermore, compensatory base changes (CBCs; nucleotide changes at both strands of the paired bases and lead to functional constraints on secondary structure) that highly informative for identifying closely related species (*Budak et al., 2016*; *Coleman, 2007*). The integration of ITS2 secondary structure and CBCs analysis has been increasingly utilised to support morphological identifications and uncover cryptic diversity (*Blaxter et al., 2005*; *Chase & Fay, 2009*).

The *COI* and ITS2 have been used as molecular identifiers in copepod studies, however, but their combined application for species delimitation within Harpacticoida has not yet been investigated. This study is the first to apply a dual-marker approach, integrating *COI* and ITS2 data, to investigate species boundaries within *Arenosetella* populations as a model. Furthermore, it represents the first instance within Harpacticoida where ITS2 secondary structure and CBCs are employed as additional features for species delimitation. Additionally, this study provides the first barcode data for both *COI* and ITS2 markers from a Harpacticoid species along the coasts of Türkiye, thus contributing to the regional and global understanding of this group's genetic diversity. While the focus remains on *Arenosetella*, the approach establishes a broader framework that can be extended to other Harpacticoida, setting a precedent for resolving taxonomic ambiguities and advancing our understanding of biodiversity within this diverse and ecologically significant order. The objective of this study is to address the cryptic diversity present within *Arenosetella* by comparing molecular operational taxonomic units (MOTUs) identified with each molecular identifier and examining potential morphological correlates. This approach underscores the significance of integrating molecular and structural data for robust species delimitation.

## MATERIALS & METHODS

### Sampling

Within the scope of the TÜBİTAK project 119Z820, surveys were conducted at 123 stations along the Mediterranean, Aegean, and Black Sea coasts of Türkiye (under the permission of the General Directorate of Nature Conservation and National Parks, Ministry of Agriculture and Forestry of the Republic of Türkiye, permit number: 21264211-288.04; Biodiversity Research Permits-E.1495402, dated 14/05/2019), with *Arenosetella* species detected at a total of nine specific localities, comprising three sites from each coastline (Table S1). Sampling took place in the intertidal zones using the Karaman-Chappius method (*Delamare-Deboutteville, 1954*), where small pits were dug in wet areas at the boundary of the wave-washed shore—specifically the zone where waves periodically recede and rewet the sand. Specimens were collected from the water within these pits, filtered through 60 $\mu$m silk nets, washed, and stored in 99% undenatured ethanol in sealed 100 cm$^3$ plastic containers. Specimens were then transferred to petri dishes and sorted using an

Olympus SZX16 stereo microscope. Sorted specimens were stored in five ml tubes with 99% ethanol at −20 °C until all sorting process was complete. Afterward, specimens were transferred to cavity slides with propylene glycol for initial morphological identification under an Olympus BX53 binocular microscope at low light intensity. The identified specimens were then preserved in 99% ethanol at −20 °C until DNA extraction.

## Generation of sequence data

Total genomic DNA was extracted from ethanol-preserved, morphologically intact specimens using a protocol based on *Easton & Thistle (2014)*. This protocol was selected because it did not damage the specimens, which were then used to subsequent morphological examination. The mitochondrial *COI* gene region was amplified under the standard and nested polymerase chain reaction (PCR) approaches using the primer pairs CoxF, CoxR2 (*Cheng et al., 2013*) and Cop-*COI* + 20 (*Chang, 2007*), HCO2198 (*Folmer & Black, 1994*), respectively (please see Table S3 to primer sequence information). The ITS2 region was amplified using the primer pair of CAS5p8sFc and CAS28sB1d (*Ji, Zhang & He, 2003*) (Table S3). Amplifications were carried out in 25-µl volumes containing 0.25 U of Taq polymerase, 2.5 µl of 10x reaction buffer (100 mM Tris–HCl, pH 8.8, 500 mM KCl and 0.8% Nonidet P-40), 10 pmol of each of the primers, 0.2 mM of each of the four dNTPs, 1.5 mM MgCl2, 0.6% DMSO and five µl of DNA template (20–50 ng). PCR cycle conditions were: 5 min at 94 °C; 40 cycles of 45 s at 94 °C, 30 s at 42.7–50.5 °C (depending on the primers used, Table S3), 60 s at 72 °C and a final extension at 72 °C for 5 min. The purified PCR products were sequenced in both directions using the same primers as in PCR reactions at Macrogen Inc. Sequences produced in this study were deposited in the GenBank database with the accession numbers of PV537515–PV537560 for *COI* and PV547651–PV547696 for ITS2 (Table S2).

## Data analysis

### Analysis of the COI gene region

The forward and reverse nucleotide sequences were assembled, edited and aligned by eye using Geneious R9 (*Kearse et al., 2012*). Three *COI* sequences of *A. germanica* (MH670488, MH670489, and MH670491) (*Rossel & Martínez Arbizu, 2019*) were retrieved from the database of NCBI GenBank. *Ectinosoma soyeri* (Harpacticoidea: Ectinosomatidae), sequenced in this study, was used as an outgroup. Multiple alignment of *COI* was performed using MAFFT v7.017 (*Katoh & Standley, 2013*) and the aligned dataset was then collapsed into haplotypes using DnaSP 5.0 (*Librado & Rozas, 2009*).

Maximum Likelihood (ML) tree was constructed in IQ-Tree (*Trifinopoulos et al., 2016*; *Hoang et al., 2018*) under the K3Pu+F+G4 model of nucleotide substitution that was inferred as the best fit model by ModelFinder (*Kalyaanamoorthy et al., 2017*; *Bouckaert et al., 2014*), with a total of 1,000 ultrafast bootstrap replicates to assess branch supports. Bayesian inference (BI) analysis of ultrametric trees was conducted in BEAST v2.0 (*Bouckaert et al., 2014*) under the GTR+I+G model. Distribution of posterior parameters was estimated in two independent runs with four Markov chains (three cold, one heated) based on 10 million generations and sampling every 1,000 generations. The log-likelihood files produced by each run were assessed considering effective sample size (ESS > 200)

for all priors using Tracer v1.7 (*Rambaut et al., 2018*). The first 25% of trees sampled in each run were then eliminated as burn-in, and a majority-rule consensus tree (BI tree) was constructed from the remaining trees. The obtained trees were then visualised in FigTree v1.4.2 (*Rambaut, 2014*).

COI-based species delimitation was performed using the Automatic Barcode Gap Discovery (ABGD) (*Puillandre et al., 2012*), Assemble Species by Automatic Partitioning (ASAP) (*Puillandre, Brouillet & Achaz, 2021*), Statistical parsimony analysis (TCS) (*Templeton, 2001*), Bayesian Poisson tree processes (bPTP) model (*Zhang et al., 2013*) and Generalized Mixed Yule Coalescent (GMYC) approach (*Fujisawa & Barraclough, 2013*). ABGD was conducted on the online platform (*Puillandre et al., 2012*) with default parameters using a Kimura 2 parameter (K80 model). ASAP was carried out using the ASAP webserver (*Puillandre, Brouillet & Achaz, 2021*) under the K80 model. TCS analysis was performed as implemented in TCS v1.2.1 (*Clement, Posada & Crandall, 2000*) with 90% and 95% connection limits to further assess species clusters. GMYC was conducted on the online GMYC server (*Fujisawa & Barraclough, 2013*; https://species.h-its.org/gmyc/) using the ultrametric tree constructed in BEAST2 with a single-threshold setting to delineate species. Finally, the PTP modelling was performed with PTP web server (*Zhang et al., 2013*; https://species.h-its.org/ptp/) under the maximum likelihood implementation (mlPTP) with a single Poisson distribution and the Bayesian implementation (bPTP) using the generated ML tree as input tree and default parameters as MCMC thinning set to 100 and a burn-in of 0.1.

## ITS2 region analysis

In order to predict secondary structures of ITS2, ITS2 sequences were firstly trimmed of flanking 5.8S and 28S proximal stem motifs using the HMM-based annotation tool present at the ITS2 database tool (*Koetschan et al., 2009*; *Selig et al., 2008*; *Schultz et al., 2006*), with default parameters. Due to the presence of noisy reads in some sequences within the 5.8S or 28S regions, boundary trimming was confirmed with reference sequences where necessary. Subsequently, the secondary structures for ITS2 sequences were predicted using the cpPredictor tool, which is based on homology and is available in the ITS2 database (*Jelínek & Pánek, 2019*; *Pánek, Modrák & Schwarz, 2017*). The values were selected as hairpin threshold (30%), stem threshold (20%) and compute z-score no. The predicted structures were then visualised with VARNA 3.9 (*Darty, Denise & Ponty, 2009*) to confirm their structural integrity. MARNA (*Siebert & Backofen, 2005*) was used to generate a multiple alignment incorporating both nucleotide sequence and secondary structure homology. CBC matrices were generated using the CBC Matrix function in the 4SALE software (*Seibel et al., 2008*), allowing for the assessment of compensatory base changes within ITS2 secondary structures (*Coleman, 2003*; *Coleman, 2007*; *Schultz et al., 2005*; *Wolf et al., 2005*).

The species delimitation process was conducted utilising the ABGD, ASAP, and bPTP analyses, as outlined in the *COI* section. For ABGD and ASAP, distance matrices that account for both nucleotide sequences and secondary structures were created using ProfDist (*Wolf et al., 2008*), employing the "General Time Reversible" and "Ratematrix Q" models for ITS2-specific corrections. The bPTP analysis employed a neighbour-joining

(NJ) tree constructed in ProfDist, with topology testing by bootstrap analysis set to 1,000 replicates (*Felsenstein, 1985*).

## Combined dataset analysis

Due to the discrepancy in the number of haplotypes observed among the *COI* and ITS2, the combined analyses were performed on individual sequences rather than on haplotypes. Sequences from outgroup and database sources were excluded, and the *COI* and ITS2 alignments were merged using SequenceMatrix v1.9 (*Vaidya, Lohman & Meier, 2011*) to generate a concatenated dataset. Subsequently, GMYC and bPTP analyses were applied in accordance with the steps previously described for *COI*.

## Testing gene tree-species tree congruence: STARBEAST

In order to test the concordance between gene trees and the species boundaries proposed by species delimitation methods, the STARBeast approach was preferred as using four candidate species due to the result of most analyses including individual and combined datasets, which supported the presence of four candidate species for two known morpho-species along the Turkish coasts. This approach was conducted using the StarBEAST2 package in BEAST v2.6.7 (*Heled & Drummond, 2010*; *Bouckaert et al., 2014*). The concatenated dataset was analysed with *COI* designated as mitochondrial (ploidy 0.5) and ITS2 as nuclear (ploidy 2.0). Simulations were run for five million generations, sampling every 1,000 generations. Effective sample size (ESS) values were assessed in TRACER v1.5 and the final species tree was visualised using DensiTree (*Bouckaert et al., 2014*).

## Morphological comparisons

Morphological analyses were conducted on 46 individuals in which the *COI* and ITS2 regions were successfully sequenced. The specimens were mounted in lactophenol for examination, following the protocol by *Karaytuğ & Sak (2006)*, to prevent collapse for high-quality morphological comparisons. Distinguishing characteristics among MOTUs were digitally drawn from microscope images using a drawing tube attached to an Olympus BX53 microscope and then processed in Adobe Photoshop 2024.

# RESULTS

The sequence information of the *COI* and *ITS2* barcode regions were generated in 46 specimens representing the species of *Arenosetella* along the Turkish coasts. All of the exoskeletons were successfully recovered for morphological assessments. The findings of the study were presented below in detail.

## *COI* barcode region

After alignment and trimming of the *COI* barcode region, the remaining length of sequences was 663 bp, with 241 variable positions, 422 conserved sites, and 84 parsimony-informative sites. The nucleotide composition biased towards A and T nucleotides, with an average 60.4% AT content (Table S4). The average of GC content varied across codon positions, being 46.3%, 42.3% and 30.4% at the first, second and third codon positions, respectively

Table 1 Haplotype distribution in *Arenosetella* populations based on the *COI* and ITS2. Haplotypes generated from the COI and ITS2 regions are labelled as Hap_ and Set_, respectively.

| Out group | Sample name *Ectinosoma aff. soyeri* | *COI* Hap_1 | ITS2 | | Sample name 15H1 | *COI* Hap_18 | ITS2 Set_7 |
|---|---|---|---|---|---|---|---|
| NCBI | MH670489 | Hap_3 | | | 15H3 | Hap_19 | Set_8 |
| | MZ343338 | Hap_2 | | | 15H5 | Hap_19 | Set_8 |
| | MH670491 | Hap_2 | | | 15H6 | Hap_18 | Set_7 |
| | MH670488 | Hap_2 | | | 15H9 | Hap_18 | Set_7 |
| | 97H4 | Hap_6 | Set_2 | | 15H10 | Hap_20 | Set_9 |
| | 97H6 | Hap_6 | Set_2 | | 15H16 | Hap_21 | Set_7 |
| | 97H7 | Hap_6 | Set_2 | | 15H17 | Hap_22 | Set_7 |
| | 97H8 | Hap_6 | Set_2 | MEDITERRENIAN SEA | 29H1 | Hap_16 | Set_6 |
| | 97H9 | Hap_6 | Set_2 | | 29H2 | Hap_16 | Set_6 |
| | 97H10 | Hap_10 | Set_2 | | 29H4 | Hap_16 | Set_6 |
| BLACK SEA | 103H21 | Hap_7 | Set_1 | | 29H5 | Hap_16 | Set_6 |
| | 103H22 | Hap_8 | Set_1 | | 29H6 | Hap_16 | Set_6 |
| | 103H23 | Hap_9 | Set_1 | | 29H7 | Hap_16 | Set_6 |
| | 103H24 | Hap_8 | Set_1 | | 29H8 | Hap_16 | Set_6 |
| | 113H12 | Hap_5 | Set_1 | | 32H1 | Hap_16 | Set_6 |
| | 113H13 | Hap_4 | Set_1 | | 32H2 | Hap_17 | Set_6 |
| | 113H15 | Hap_5 | Set_1 | | 32H3 | Hap_16 | Set_6 |
| | 113H27 | Hap_6 | Set_1 | | | | |
| | 46H1 | Hap_15 | Set_5 | | | | |
| | 46H2 | Hap_15 | Set_5 | | | | |
| | 46H3 | Hap_15 | Set_5 | | | | |
| | 46H5 | Hap_15 | Set_5 | | | | |
| | 46H6 | Hap_15 | Set_5 | | | | |
| | 61H1 | Hap_13 | Set_4 | | | | |
| AGEAN SEA | 61H2 | Hap_13 | Set_4 | | | | |
| | 61H3 | Hap_13 | Set_4 | | | | |
| | 61H9 | Hap_13 | Set_4 | | | | |
| | 61H11 | Hap_14 | Set_4 | | | | |
| | 61H12 | Hap_13 | Set_4 | | | | |
| | 78H1 | Hap_11 | Set_3 | | | | |
| | 78H2 | Hap_12 | Set_3 | | | | |
| | 78H3 | Hap_11 | Set_3 | | | | |

(see Table S4). The *COI* region yielded a total of 21 haplotypes with two sequences retrieved from the GenBank database (North Sea specimens) (Table 1). The phylogenetic analyses have recovered the same tree topologies with high support values (Fig. 1). The recovered trees formed two main clades corresponding to two morphological species, *A. germanica* and *A. lanceorostrata*, with a clear biogeographic pattern observed for *A. germanica*. The retrieved specimens from the GenBank database (North Sea) grouped as a sister clade to the Mediterranean populations of *A. germanica* (Fig. 1).

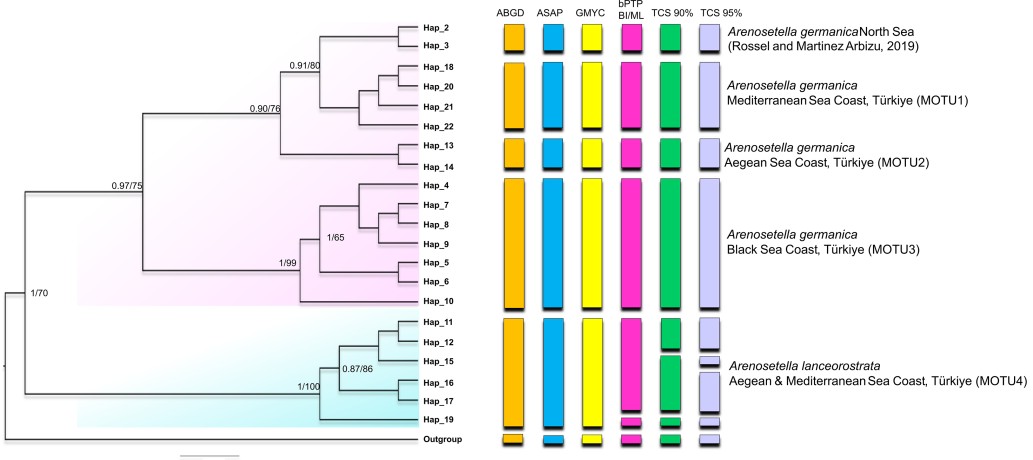

**Figure 1** **Phylogenetic tree generated from the *COI* haplotypes using Bayesian inference (BI) and maximum likelihood (ML) methods.** Support values are shown on the tree. The obtained COI haplotypes are abbreviated as Hap. The outgroup (*Ectynosoma aff. soyeri*) is shown as Hap_1. The vertical bars represent the taxonomic units (MOTUs) identified by the species delimitation analyses (ABGD, Automatic Barcode Gap Discovery; ASAP, Assemble Species by Automatic Partitioning; GMYC, Generalized Mixed Yule Coalescent; bPTP, Bayesian Poisson tree processes; TCS, Statistical parsimony analysis). Coloured backgrounds refer to two morphological species, *A. germanica* (pink lace) and *A. lanceorostrata* (cyan).

The result of the species delimitation analyses, incorporating North Sea sequences of *A. germanica* and the outgroup, was visually summarised by vertical colour bars on the right side of the phylogeny in Fig. 1. The North Sea sequences (Hap_2 and Hap_3), attributed to *A. germanica*, consistently represented a distinct MOTU across all delimitation analyses, highlighting their genetic distinctiveness. The ABGD, ASAP and GMYC analyses indicated the presence of four MOTUs among the Turkish populations, while only slight variation in the composition of MOTU4 observed in the bPTP and TCS analyses (Fig. 1 and Figs. S1–S3). Three of the identified MOTUs corresponded to *A. germanica* populations, consistent with their geographic distribution: MOTU1 (Mediterranean), MOTU2 (Aegean) and MOTU3 (Black Sea). The populations of *A. lanceorostrata* were represented by a single MOTU (MOTU4) encompassing specimens from both the Aegean and Mediterranean coasts (Fig. 1). In the bPTP analysis, MOTU4 (Hap_11-12, Hap_15-17, Hap_19) forming the *A. lanceorostrata* populations was subdivided into two groups: MOTU4a (Hap_11-12, Hap_15-17) and MOTU4b (Hap_19) (Fig. S3). A similar pattern was observed in the TCS network at a 90% connection limit (Fig. 1), with the exception of MOTU4a (Fig. S3), which was further split into two groups—Hap_11-12 (Aegean) and Hap_15-17 (Aegean and Mediterranean). At the 95% connection limit, Hap_15 (Aegean), was resolved as a distinct subgroup indicating additional genetic structure within *A. lanceorostrata*.

## ITS2 region

The GC content of the ITS2 sequences ranged from 55.7 to 62.3%, with a 60.1% on average (Table S5). The ITS2 sequences displayed variation in length ranging from 226 bp and

234 bp, with the presence of substitutions and/or indels (Table S5, Fig. S4). The ITS2 region yielded a total of nine haplotypes with two from the Black Sea, three from the Aegean, and four from the Mediterranean (Table 1). The alignment of ITS2 sequences for secondary structure comparisons in MARNA generated 250 positions in length with 24 indels (Fig. S4). The ITS2 secondary structures were shown for each haplotype in Fig. S5. Notwithstanding the variable nucleotide positions, the analysis of the ITS2 folding pattern of all samples yielded two secondary structures that were broadly similar, one consisting of three helices and the other of four. The first predicted secondary structure comprising three helices (helix I, II, and III) was observed in the Aegean and Mediterranean populations of *A. germanica* (Set_4, Set_7 and Set_9), while the second structure comprising four helices with the presence of helix IV was specific to Black Sea populations of *A. germanica* (Set_1 and Set_2) and all populations of *A. lanceostrata* (Set_3, 5, 6 and 8). Furthermore, this second structure exhibited a discrepancy with an additional helix (Helix IIA) (Fig. S5). The homologous segments of the predicted structures were found to be in homologous locations. Helix I formed a non-dichotomous structure with a variable length between 31 bp (Set_4) and 37 bp (Set_7, Set_9) (Fig. 2 and Table S6). Helix II was the most conserved, featuring a pyrimidine-pyrimidine bulge and a non-canonical U(T)-G base pair, with a highly conserved motif (5′GCUCUCGCGGAGUGAAAUCCGCGUGGC) (Fig. 2). Helix III is the longest, ranging from 89 bp in all *A. lanceostrata* specimens (Set_3, 5, 6, 8) to 157 bp in the specimen of *A. germanica* (Set_7, Set_9) (Table S6). This helix produced two distinct folding patterns with a branched structure following a single helix in the first predicted (with three helices) and with a non-dichotomous structure in the second predicted (with four helices) (Fig. S5). This helix also included an individually recognizable motif (5′AUCCUCCGGGAA) in relatively close location to its the 5′ apex (Fig. 2). Additionally, Helix IV was the shortest with 31 bp and 32 bp in length (Table S6).

The result of the species delimitation analyses using the ITS2 region are represented by vertical colour bars on the right side of the constructed phylogenetic tree in Fig. 3. All subsequent delimitation analyses consistently revealed the presence of four MOTUs. The initial three MOTUs (MOTU1–MOTU3) encompassed the all populations of *A. germanica*, with the occurrence of a relation with their distribution patterns as follows; MOTU1 from the Mediterranean, MOTU2 from the Aegean and MOTU3 from the Black Sea. The last MOTU (MOTU4) was represented solely by *A. lanceorostrata* populations (Fig. 3).

Nucleotide similarities within the MOTUs representing *A. germanica* have been observed to range from 90.0% (MOTU3) to 93.2% (MOTU1), while the low level was observed among these MOTUs, with an average of 75.6%. A comparable pattern was also found for MOTU4 (*A. lanceorostrata*), exhibiting an average similarity of 89.2%. However, the nucleotide similarity was the lowest between the MOTUs representing *A. germanica* and *A. lanceorostrata*, with an average of 66.4%.

The CBC analysis of the aligned putative ITS2 secondary structures indicated the presence of six CBCs in total; four in helix I and one in helix III and one in helix IV among four MOTUS (Table 2). Within *A. germanica*, the CBCs were not detected between MOTU1 and MOTU2, or between MOTU2 and MOTU3. However, two CBCs (H1_CBC3 and H1_CBC4) were identified between MOTU1 and MOTU 3. Furthermore, the occurrence

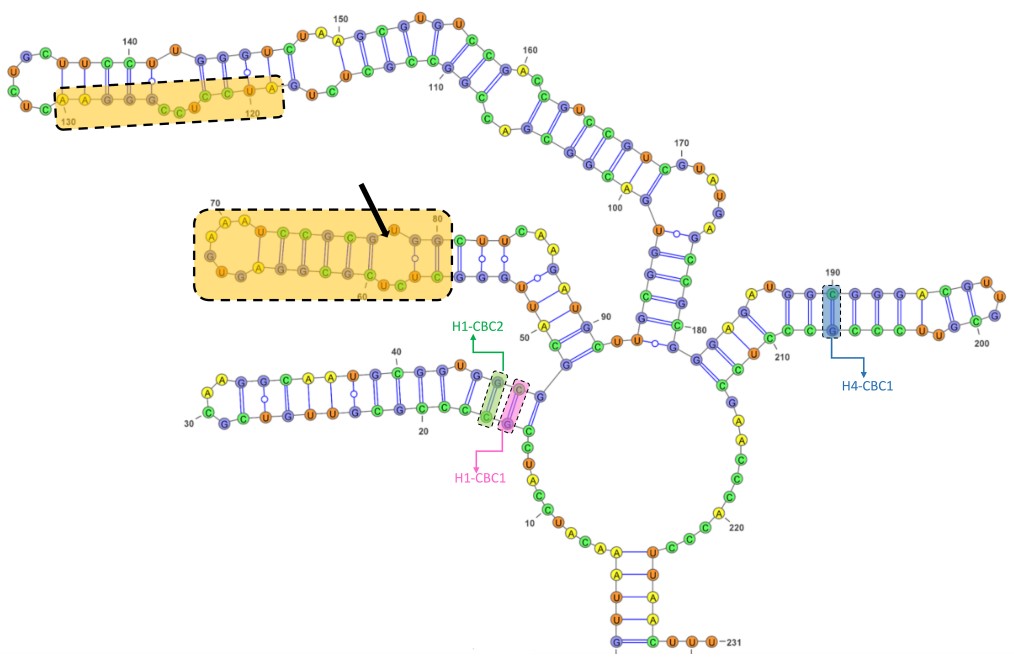

**Figure 2  The predicted structures of the ITS2 helices of *Arenosetella*.** The location and sequences of conserved motifs are highlighted by light orange colour and the pyrimidine-pyrimidine mismatch in the Helix II is indicated by arrow. The identified CBCs were shown on the helices. H1-CBC1 and H1-CBC2 indicate the related CBC on the helix I, while H4-CBC1 indicates the related CBC on the helix IV).

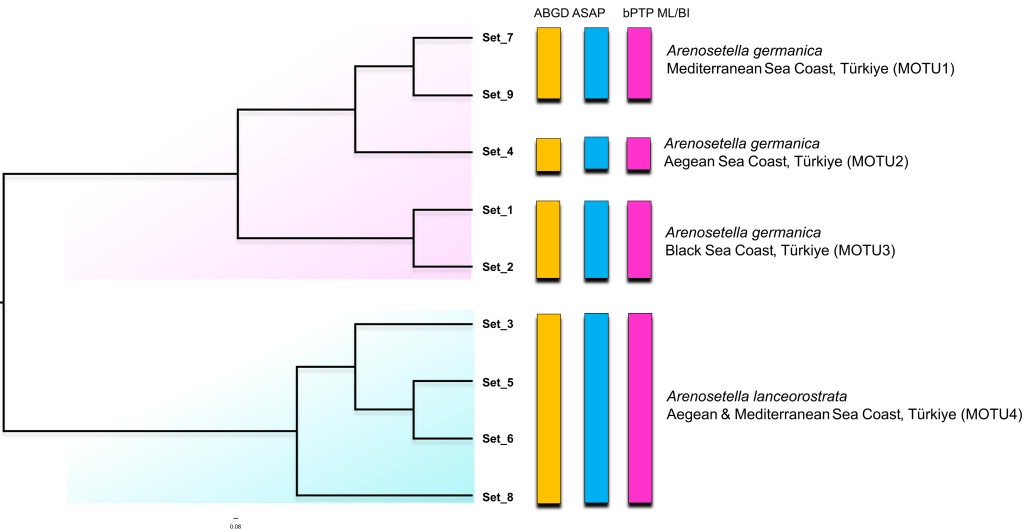

**Figure 3  Phylogenetic tree generated from the *ITS2* haplotypes using the neighbor-joining tree constructed in ProfDist (PNJ) approach.** The obtained ITS2 haplotypes are abbreviated as Set. The vertical bars represent the molecular taxonomic units (MOTUs) identified by the species delimitation analyses (please see Fig. 1 for abbreviations). Coloured backgrounds refer to two morphological species, *A. germanica* (pink lace) and *A. lanceorostrata* (cyan).

of four distinct CBCs (H1_CBC1, H1_CBC2, H3_CBC1 and H4_CBC1; Table 2) was found between MOTU4 (*A. lanceorostrata*) and the remaining three MOTUs.

## Combined dataset analyses

The analyses on the combined dataset supported a four-candidate species hypothesis under bPTP with ML and BI approaches (Fig. 4), revealing a fine-scale geographic structuring in *A. germanica* populations (Mediterranean = MOTU1, Aegean = MOTU2, Black Sea = MOTU3), as observed in single-marker analyses (Figs. 1 and 2). In addition, the populations of *A. lanceorostrata* were grouped as a single MOTU (MOTU 4). However, the GMYC analysis revealed further substructure within the populations of *A. germanica*, with the exception of MOTU2. Here, each of MOTU1 and MOTU3 was divided into two groups. In this analysis, *A. lanceorostrata* was also further divided into four distinct groups, representing the Aegean and Mediterranean coasts. The STARBeast approach for the further investigation of species boundaries supported the presence of four-candidate species, with high posterior probability (PP = 1.00) for all nodes, highlighting a robust agreement between the proposed species hypothesis and the combined dataset (Fig. 5).

## Morphological comparisons

Species delimitation analyses under multiple approaches suggested up to seven candidate species along the Turkish coasts (Figs. 1, 2 and 5). Based on these results, exoskeletons were categorized into seven groups, and morphological characters were compared within and between these groups. No significant morphological differences were identified among MOTU4–MOTU7 (*A. lanceorostrata*) in the seven-group model. However, subtle but consistent differences across MOTUs were observed in anal somite ornamentation and female P5 shape for *A. germanica* (MOTU1–MOTU3) (Fig. 6). In particular, examination of the female P5 revealed notable distinctions in the origin and proportions of the outermost setae. When the four terminal elements of the P5 exopod were numbered from inner to outer, the third and fourth setae (seta 3 and seta 4) exhibited clear structural variation these two setae arise from a well-defined common lobe in both MOTU1 and MOTU2, whereas this lobe was markedly reduced in MOTU3 (arrowed in Fig. 6A). Moreover, the relative lengths of these setae varied among MOTUs: seta 4 is approximately half the length of seta 3 in MOTU2 and MOTU3; in contrast, seta 3 was only slightly shorter than seta 4 in MOTU1. Additional differences were observed in the dorsal ornamentation of the anal somite, particularly in the length ratio of the inner and outer claw-like projections. The inner projection was slightly longer than the outer one in MOTU3, the outer was longer in MOTU2; and both projections are nearly equal in length in MOTU1.

## DISCUSSION

### Evidence for potentially distinct lineages within *A. germanica* and Monophyly of *A. lanceorostrata*

*Arenosetella germanica*, first described by Kunz (1937) from Kiel Bay, has since been recorded across multiple biogeographic regions, displaying minor but remarkable morphological variations. Although Kunz's initial description did not indicate intraspecific

Bakmaz et al. (2025), *PeerJ*, DOI 10.7717/peerj.19870

**Table 2 Nucleotide positions and occurrence of CBCs on the predicted secondary structure of ITS2 in *Arenosetella*.** Haplotypes generated from the ITS2 region are labelled as Set_. The molecular taxonomic unit is abbreviated as MOTU. The numbers in square brackets [ ] indicate the positions on the secondary structure. The letters in square brackets represent the nucleotides at those positions (*e.g.*, [15–47] [A–U] means that nucleotide A at position 15 is bonded to nucleotide U at position 47). The identified CBCs are indicated using symbols: H1_CBC1 (‡), H1_CBC2 (#), H1_CBC3 (*), H1_CBC4 (§), H3_CBC1 (◊), H4_CBC1 (Δ).

| MOTU | | Set_1 | Set_2 | Set_4 | Set_7 | Set_9 | Set_3 | Set_5 | Set_6 | Set_8 |
|---|---|---|---|---|---|---|---|---|---|---|
| MOTU3 | Set_1 | − | 0 | 0 | [19–43] [G-C]* <br> [23–40] [A-U] § | [19–43] [G-C] * <br> [23–40] [A-U] § | [16–46] [A-U] ‡ <br> [17–45] [G-C] # <br> [204–221] [G-C] Δ | [16–46] [A-U] ‡ <br> [17–45] [G-C] # <br> [204–221] [G-C] Δ | [17–45] [G-C] # <br> [204–221] [G-C] Δ | [16–46] [A-U] ‡ <br> [17–45] [G-C] # <br> [204–221] [G-C] Δ |
| MOTU3 | Set_2 | 0 | − | 0 | [19–43] [G-C] * <br> [23–40] [A-U] § | [19–43] [G-C] * <br> [23–40] [A-U] § | [16–46] [A-U] ‡ <br> [17–45] [G-C] # <br> [204–221] [G-C] Δ | [16–46] [A-U] ‡ <br> [17–45] [G-C] # <br> [204–221] [G-C] Δ | [17–45] [G-C] # <br> [204–221] [G-C] Δ | [16–46] [A-U] ‡ <br> [17–45] [G-C] # <br> [204–221] [G-C] Δ |
| MOTU2 | Set_4 | 0 | 0 | − | 0 | 0 | [16–46] [A-U] ‡ <br> [17–45] [G-C] # <br> [204–221] [G-C] Δ | [16–46] [A-U] ‡ <br> [17–45] [G-C] # <br> [204–221] [G-C] Δ | [17–45] [G-C] # <br> [204–221] [G-C] Δ | [16–46] [A-U] ‡ <br> [17–45] [G-C] # <br> [204–221] [G-C] Δ |
| MOTU1 | Set_7 | [19–43] [A-U] * <br> [23–40] [G-C] § | [19–43] [A-U] * <br> [23–40] [G-C] § | 0 | − | 0 | [16–46] [A-U] ‡ <br> [17–45] [G-C] # <br> [105–170] [C-G] ◊ | [16–46] [A-U] ‡ <br> [17–45] [G-C] # <br> [105–170] [C-G] ◊ | [17–45] [G-C] # <br> [105–170] [C-G] ◊ | [16–46] [A-U] ‡ <br> [17–45] [G-C] # <br> [105–170] [C-G] ◊ |
| MOTU1 | Set_9 | [19–43] [A-U] * <br> [23–40] [G-C] § | [19–43] [A-U] * <br> [23–40] [G-C] § | 0 | 0 | − | [16–46] [A-U] ‡ <br> [17–45] [G-C] # <br> [105–170] [C-G] ◊ | [16–46] [A-U] ‡ <br> [17–45] [G-C] # <br> [105–170] [C-G] ◊ | [17–45] [G-C] <br> [105–170] [C-G] ◊ | [16–46] [A-U] ‡ <br> [17–45] [G-C] # <br> [105–170] [C-G] ◊ |
| MOTU4 | Set_3 | [16–46] [G-C] ‡ <br> [17–45] [C-G] # <br> [204–221] [C-G] Δ | [16–46] [G-C] ‡ <br> [17–45] [C-G] # <br> [204–221] [C-G] Δ | [16–46] [G-C] ‡ <br> [17–45] [C-G] # <br> [204–221] [C-G] Δ | [16–46] [G-C] ‡ <br> [17–45] [C-G] # <br> [105–170] [G-C] ◊ | [16–46] [G-C] ‡ <br> [17–45] [C-G] # <br> [105–170] [G-C] ◊ | − | 0 | 0 | 0 |
| MOTU4 | Set_5 | [16–46] [G-C] ‡ <br> [17–45] [C-G] # <br> [204–221] [C-G] Δ | [16–46] [G-C] ‡ <br> [17–45] [C-G] # <br> [204–221] [C-G] Δ | [16–46] [G-C] ‡ <br> [17–45] [C-G] # <br> [204–221] [C-G] Δ | [16–46] [G-C] ‡ <br> [17–45] [C-G] # <br> [105–170] [G-C] ◊ | [16–46] [G-C] ‡ <br> [17–45] [C-G] # <br> [105–170] [G-C] ◊ | 0 | − | 0 | 0 |
| MOTU4 | Set_6 | [17–45] [C-G] # <br> [204–221] [C-G] Δ | [17–45] [C-G] # <br> [204–221] [C-G] Δ | [17–45] [C-G] # <br> [204–221] [C-G] Δ | [17–45] [C-G] # <br> [105–170] [G-C] ◊ | [17–45] [C-G] # <br> [105–170] [G-C] ◊ | 0 | 0 | − | 0 |
| MOTU4 | Set_8 | [16–46] [G-C] ‡ <br> [17–45] [C-G] # <br> [204–221] [C-G] Δ | [16–46] [G-C] ‡ <br> [17–45] [C-G] # <br> [204–221] [C-G] Δ | [16–46] [G-C] ‡ <br> [17–45] [C-G] # <br> [204–221] [C-G] Δ | [16–46] [G-C] ‡ <br> [17–45] [C-G] # <br> [105–170] [G-C] ◊ | [16–46] [G-C] ‡ <br> [17–45] [C-G] # <br> [105–170] [G-C] ◊ | 0 | 0 | 0 | − |

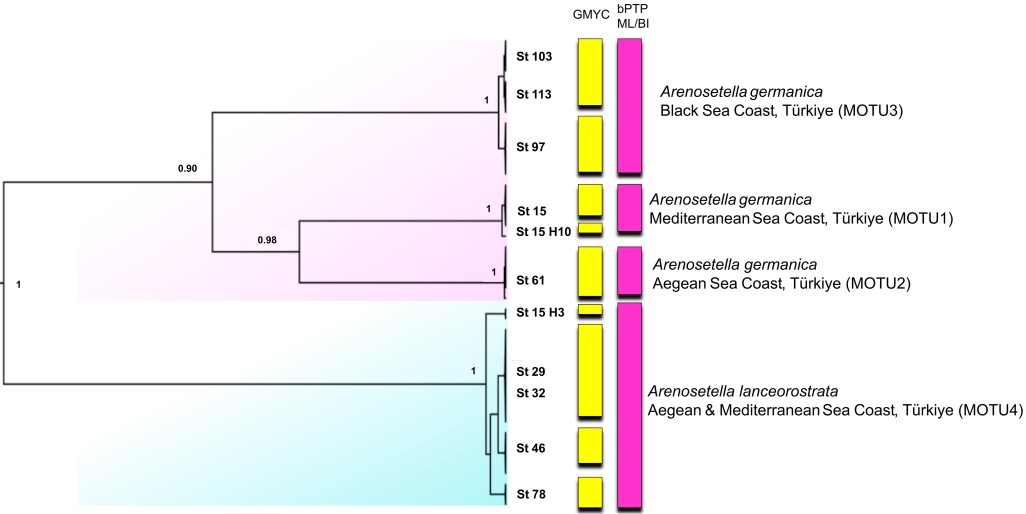

**Figure 4** **Phylogenetic tree generated from the concatenated dataset (*COI* and ITS2) of all specimens using Bayesian inference (BI) method.** Support values are shown on the tree. St refers to the sampled stations (please see Table S1). The vertical bars represent the molecular taxonomic units (MOTUs) identified by the GMYC (Generalized Mixed Yule Coalescent; coloured by yellow) and bPTP (Bayesian Poisson tree processes; coloured by pink) analyses for species delimitation. Coloured backgrounds refer to two morphological species, *A. germanica* (pink lace) and *A. lanceorostrata* (cyan).

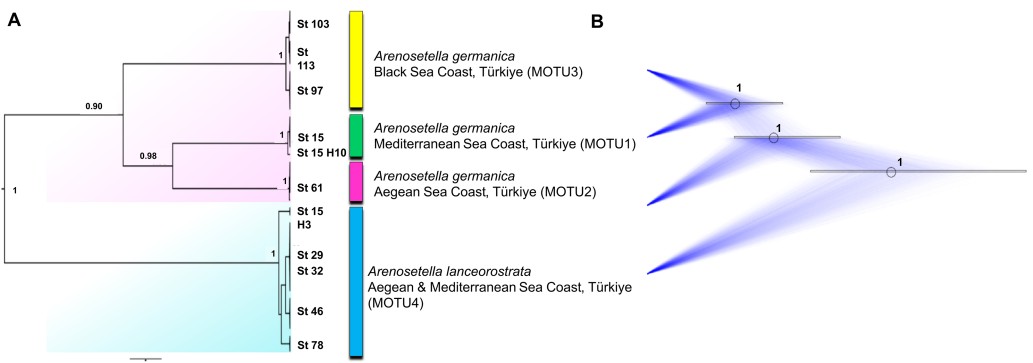

**Figure 5** **Summary scheme of the species delimitation results.** The Densitree represents 5,000 trees inferred with StarBEAST2 package in BEAST v2.6.7 based on the concatenated dataset. The posterior probability of each node is labelled with the numbers.

variability, later reports by *Chappius (1954)* in Madagascar and *Rouch (1962)* in Brazil noted differences in P3 and P4 setal formulas, interpreted as intraspecific variation. Subsequently, *Lang (1965)* re-evaluated this variability and described two new species, *A. madagascariensis* and *A. rouchi*. Despite these revisions, records of *A. germanica* continue to expand in morphological variability, suggesting the possibility of a species complex.

The findings from the Turkish coasts, combining morphological and molecular data, reveal subtle differences in P5 structure and anal somite ornamentation among populations of *A. germanica* (Fig. 6). While these findings are preliminary, they provide strong evidence

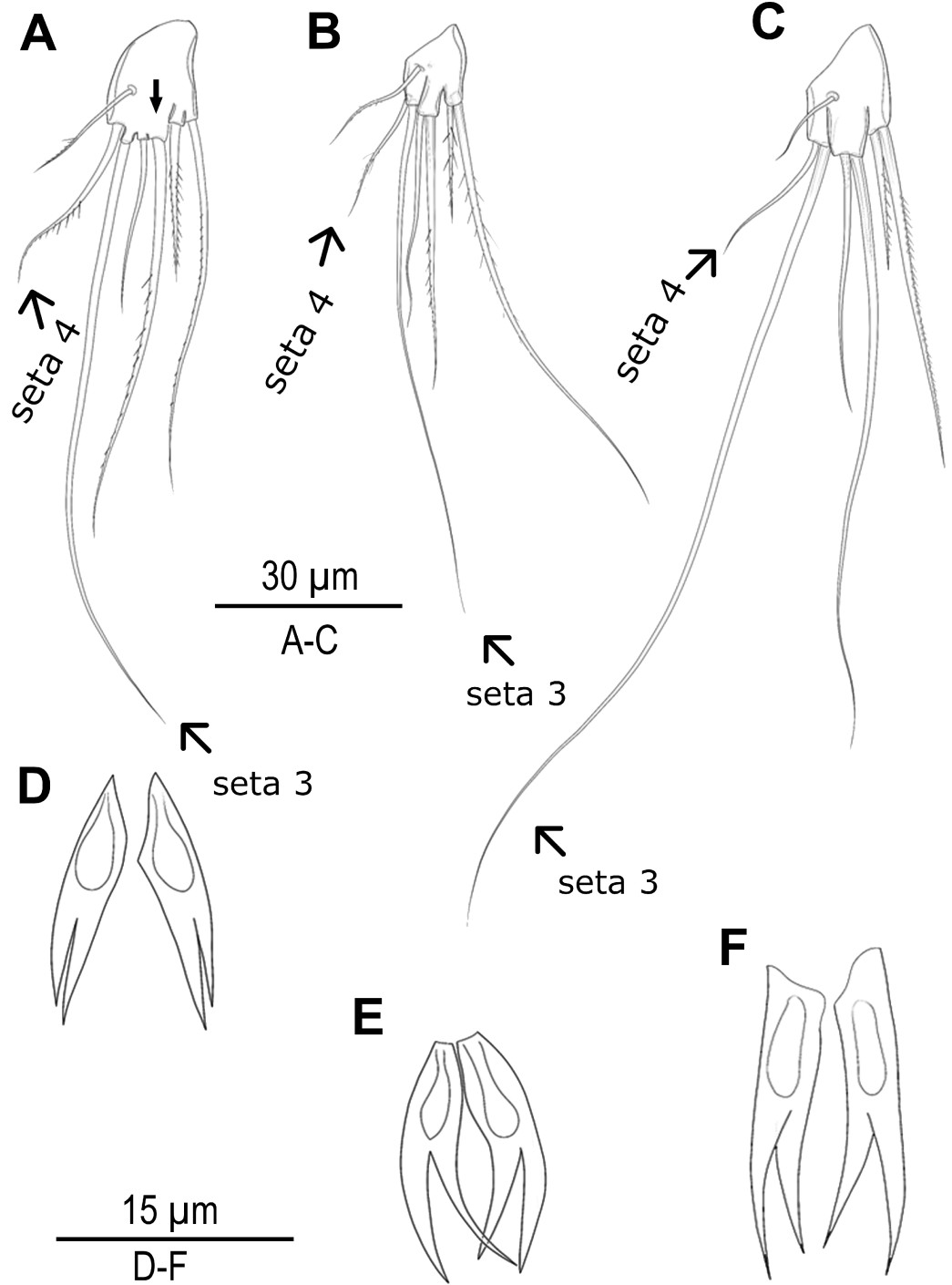

**Figure 6 Morphological variations in *Arenosetella germanica*.** (A–C) P5, anterior, (D–E) anal somite dorsal ornamentation. (A) MOTU3. (B) MOTU1. (C) MOTU2. (D) MOTU3. (E) MOTU1. (F) MOTU2.

that historical records of *A. germanica* may include multiple and unresolved lineages. This study has confirmed that two morphologically similar species, MOTU1 of *A. germanica* and *A. lanceorostrata*, which are distinguishable by subtle morphological differences and supported as separate species genetically, can coexist sympatrically (Figs. 1, 3 and 5). This perspective suggests that *A. germanica sensu Mielke (1975)* may encompass two distinct species, which were likely interpreted as intraspecific variation due to their sympatric occurrence. However, beyond Mielke's observations, all historical records of *A. germanica* warrant detailed re-evaluation to define whether they represent a species complex or intraspecific variability across its distribution range.

Geographic isolation and molecular data from the Turkish populations designate three distinct lineages—MOTU1 (Mediterranean coasts), MOTU2 (Aegean coasts), and MOTU3 (Black Sea coasts) (Figs. 1, 3 and 5)—that could represent early stages of speciation. Unlike the sympatric coexistence observed in *Mielke*'s (*1975*) records, these MOTUs do not overlap in their distribution ranges, complicating the determination of their taxonomic status.

The historical expansion of *A. germanica*'s boundaries and its tendency to incorporate morphological variants have created significant challenges for delineating species within this group. The recognition of *A. lanceorostrata* as a distinct species required robust justification. Within this context, the status of *A. lanceorostrata* as a distinct species might be questioned, particularly when considering the sympatric occurrence of these two species. While sympatry can be interpreted as evidence supporting their status as distinct species due to the lack of gene flow, it could also raise doubts about whether the observed differences merely reflect intraspecific variation within an increasingly broad species concept of *A. germanica*. Historically, the accumulation of morphological variation within *A. germanica* as seen in examples such as *A. germanica galapagoensis* and other historical records—has often led to the interpretation of such traits as intraspecific variability rather than evidence of speciation. Without molecular evidence, this ambiguity could persist, necessitating further evaluation of the relationship between these two taxa.

The sympatric occurrence of closely related species, as observed between *A. germanica* and *A. lanceorostrata*, is not an isolated phenomenon but rather appears to reflect a broader pattern characteristic of marine harpacticoids along the Turkish coasts. Studies on harpacticoid copepods in this region have revealed similar instances of congeneric coexistence. For example, faunistic surveys in the Biga Peninsula documented the sympatric presence of *Heterolaophonte brevipes* and *H. uncinata* within the same phytal samples (*Kabaca, Sak & Alper, 2022*). Likewise, *Phyllopodopsyllus briani* Petkovski, 1955 and *P. thiebaudi* Petkovski, 1955 were reported to occur sympatrically along the Marmara Sea coast of Türkiye (*Karaytuğ & Sak, 2006*).

In addition to these faunistic records, recently described harpacticoid species *Ameira venthami* and *Ameira parvula*, which differ only in subtle morphological characters, also appear to co-occur sympatrically (*Yıldız & Karaytuğ, 2024*). Although this study provides a detailed morphological assessment, it lacks molecular data, leaving open the possibility that the observed differences may represent intraspecific variation rather than true species-level divergence.

The recurrence of such patterns across diverse copepod taxa suggests that the heterogeneous coastal environments of Türkiye—with their mosaic of rocky shores, seagrass beds, and varied sediment types—offer numerous microhabitats that can facilitate both the evolution and coexistence of closely related species. Stable sympatric coexistence, however, typically requires mechanisms that reduce interspecific competition and maintain reproductive isolation. Among benthic copepods, ecological segregation may occur at fine spatial scales, including vertical partitioning within sediment layers or selective association with biogenic structures formed by macroalgae and other ecosystem engineers (*Sbrocca et al., 2021*).

From an evolutionary perspective, the persistence of reproductive boundaries in sympatry often relies on robust prezygotic isolation mechanisms. In copepods, mate recognition is commonly mediated by species-specific pheromonal cues (*Powers et al., 2020*), while mechanical isolation—due to divergence in the morphology of reproductive structures—can prevent interspecific mating (*Ohtsuka & Huys, 2001*). In this context, the subtle but consistent differences observed in the female P5 structure among *A. germanica* lineages and *A. lanceorostrata* in our study may function not merely as diagnostic characters, but as components of a lock-and-key mechanism that reinforces reproductive isolation and preserves lineage integrity.

This study provides robust evidence to support the recognition of *A. lanceorostrata* as a distinct species (Figs. 1, 3 and 5; Table 2). First, the sympatric occurrence of *A. germanica* and *A. lanceorostrata* in the same regions strongly suggests a non-inclusive relationship, as sympatry without gene flow is a hallmark of species boundaries. Multiple genetic analyses confirm the absence of gene flow between these two taxa, providing clear evidence that their sympatry does not reflect intraspecific variation but rather supports their status as distinct species. The *COI* and ITS2 sequences reliably distinguish these two taxa, with phylogenetic analyses supporting well-resolved and distinct clades for *A. lanceorostrata*. While GMYC analysis (with concatenated dataset) suggested potential subdivisions within *A. lanceorostrata* (Fig. 4), this result likely reflects the method's tendency to overestimate species boundaries when analysing loci with differing substitution rates (*Luo et al., 2018*).

## Evaluation of dual barcoding in harpacticoid copepods

The use of *COI* and ITS2 markers allowed for a more refined species delimitation within *Arenosetella*, each marker offering distinct advantages. ITS2 amplified consistently across samples which can be due to the conserved nature of primers targeting the 5.8S and 28S regions flanking ITS2, making it easier to analyse across populations. *COI* amplification, however, was more challenging, likely due to sequence variability at primer binding sites. This required nested PCR for consistent amplification, adding complexity but allowing for broader comparison due to *COI*'s representation in databases like BOLD and GenBank. The scarcity of ITS2 sequences for Harpacticoida, however, limits comparative opportunities with other studies. In terms of analytical complexity, *COI* can be processed using standard methods, whereas ITS2 requires consideration of secondary structure. CBC analyses based on ITS2 structure highlighted conserved structural patterns that may relate to genetic isolation, adding a layer of taxonomic resolution beyond *COI*.

The analyses of ITS2 revealed structural differences and CBCs, supporting genetic distinction among candidate species. CBCs are not definitive indicators of separate species but are useful markers of potential genetic isolation. Studies on groundwater amphipods (*Kornobis & Pálsson, 2013*) and other crustaceans indicate that ITS2 structural variation aids species differentiation, an approach repeated in our findings for *Arenosetella* (Table 2). The ITS2 analyses align with those on *Daphnia longispina* (*Zuykova, 2019*), supporting species delineation through genetic isolation.

This study is the first to apply both *COI* and ITS2 with CBC analysis in Harpacticoida copepods, offering a pioneering example. Combining CBC and ITS2 structural data with traditional *COI* barcoding enhances taxonomic insights and reveals cryptic diversity in *Arenosetella*. The complementary nature of these markers—*COI*'s extensive database representation and ITS2′s structural detail—offers a balanced framework adaptable to similar taxa.

Our findings align with previous studies on the utility of ITS2 in cryptic species identification, as demonstrated in the *Daphnia longispina* complex (*Zuykova, 2019*). The sequence-structure analysis for ITS2, combined with CBCs, reinforces genetic boundaries and supports lineage differentiation. Integrating *COI* and ITS2, as applied in the *Paracalanus parvus* complex (*Cornils & Held, 2014*), provides consistent support for geographic structuring and monophyletic clades within *Arenosetella* MOTUs. The observed genetic distances and monophyly confirm patterns of genetic diversity in copepods like *Chydorus sphaericus* (*Belyaeva & Taylor, 2009*), reinforcing ITS2′s value for identifying cryptic species.

Across the Turkish coasts, molecular analysis of *COI* and ITS2 revealed genetic variability among populations, particularly within *A. germanica*. *COI* sequences revealed greater haplotype diversity, with distinct haplotype groups in the Black Sea suggesting regional divergence. Phylogenetic analyses consistently placed *A. germanica* and *A. lanceorostrata* into separate clades, with *COI* sequences from the North Sea forming a sister clade to the Turkish Mediterranean populations, suggesting biogeographic structuring and regional connectivity.

ITS2 sequences displayed greater conservation but with significant structural variation across haplotypes. High GC content and conserved secondary motifs in ITS2 suggest evolutionary stability. Secondary structure analyses showed CBCs in specific haplotypes, particularly Set_4, which exhibited unique structural divergence, supporting ITS2′s utility as a marker for identifying cryptic lineages. This approach, using CBCs and secondary structure, adds robustness to the identification of potential cryptic species.

### Future directions and taxonomic recommendations

This study highlights the need for a thorough taxonomic reassessment within *Arenosetella*, as genetic evidence suggests early-stage speciation in certain populations under investigation. The clear geographic structuring observed in *A. germanica* populations indicates the presence of distinct lineages that warrant formal taxonomic examination. While these lineages are not entirely cryptic, the morphological differences, such as the shape of the anal somite dorsal ornamentation and the structure of P5, are subtle and challenging to

observe under standard light microscopy. For instance, the anal somite ornamentation often curves dorsally to ventrally between the furcal rami, making it difficult to observe clearly due to the depth of field and occasional coverage by a pseudooperculum. Similarly, the P5 structure is often obscured by overlapping swimming legs and its boundaries are challenging to define without dissection.

*Stupnikova & Neretina (2022)* demonstrated the occurrence of mitonuclear discordance in calanoid copepods, highlighting challenges such as mitochondrial introgression and the presence of NUMTs (nuclear mitochondrial pseudogenes). These findings emphasize the importance of using both mitochondrial and nuclear markers to address such issues. By combining complementary markers, this dual-marker approach enhances the resolution of MOTUs within *Arenosetella* and provides a more reliable framework for delineating species boundaries, reducing the risk of misinterpretation caused by relying solely on mitochondrial data.

## CONCLUSIONS

In conclusion, dual barcoding with *COI* and ITS2 enhances species delimitation within *Arenosetella* and underscores the value of combining marker-specific advantages. The secondary structure analysis for ITS2 adds specificity to taxonomic assessments, while *COI* enables broader phylogenetic comparisons across taxa. This integrated approach provides a representative case for future studies within Harpacticoida, highlighting the potential of ITS2 secondary structure data in DNA barcoding and improving the accuracy of taxonomic assessments. Our findings confirm that *A. lanceorostrata* and *A. germanica* are distinct species, which co-occur sympatrically and exhibit consistent differences in setal formulae, P5 morphology and anal somite ornamentation, all supported by molecular divergence. This finding contrasts with *Mielke*'s (*1975*) interpretation, who reported sympatric *A. germanica* morphotypes with differences in setal formulae which he interpreted as intraspecific variation. Our results suggest that such cases may in fact reflect overlooked species-level divergence. However, for the geographically structured MOTUs within *A. germanica* along the Turkish coasts, we consider it premature to assign formal species status as the observed morphological differences are subtle.

Further efforts to expand ITS2 sequence databases will enhance the comparative potential of dual barcoding, particularly for copepod biodiversity studies, and provide clarity on the candidate species status suggested by our genetic data. Overall, this study supports a comprehensive approach that combines molecular and morphological evidence to refine species boundaries and advance taxonomic research in copepods.

## ACKNOWLEDGEMENTS

Support in fieldwork and sample collection was kindly provided by Prof. Dr. Süphan KARAYTUĞ (Mersin University), Prof. Dr. Serdar SAK (Balıkesir University), and Assoc. Prof. Dr. Alp ALPER (Balıkesir University), for which appreciation is extended. We thank three anonymous reviewers for their careful reading of the manuscript and many constructive comments. We acknowledge the use of Google Gemini for its assistance

in reviewing the English text of this manuscript. The tool was specifically utilized for grammatical error checking and proofreading to enhance the linguistic quality and clarity of the writing.

### Funding
This research was funded by the Scientific and Technological Research Council of Turkey (TÜBİTAK), project number 119Z820 (3501 Research Program), and by the Cumhuriyet University Scientific Research Projects Coordination Unit (CÜBAP), project number F-2022-655. The TÜBİTAK 119Z820 project provided financial support for field sampling, genomic DNA extraction, COI gene amplification, and subsequent Sanger sequencing. The amplification and Sanger sequencing of the ITS2 region were financially supported by the CÜBAP project F-2022-655. The funders had no role in study design, data collection and analysis, decision to publish, or preparation of the manuscript.

### Grant Disclosures
The following grant information was disclosed by the authors:
The Scientific and Technological Research Council of Turkey (TÜBİTAK), project number 119Z820 (3501 Research Program).
The Cumhuriyet University Scientific Research Projects Coordination Unit (CÜBAP), project number F-2022-655.
The TÜBİTAK 119Z820 project provided financial support for field sampling, genomic DNA extraction, COI gene amplification, and subsequent Sanger sequencing.
The amplification and Sanger sequencing of the ITS2 region were financially supported by the CÜBAP project F-2022-655.

### Competing Interests
The authors declare there are no competing interests.

### Author Contributions
- Dilara Bakmaz conceived and designed the experiments, performed the experiments, analyzed the data, prepared figures and/or tables, authored or reviewed drafts of the article, and approved the final draft.
- Serdar Sönmez conceived and designed the experiments, performed the experiments, analyzed the data, prepared figures and/or tables, authored or reviewed drafts of the article, and approved the final draft.
- Ertan Mahir Korkmaz conceived and designed the experiments, analyzed the data, authored or reviewed drafts of the article, and approved the final draft.

### Field Study Permissions
The following information was supplied relating to field study approvals (*i.e.*, approving body and any reference numbers):

Field sampling was conducted under the permission of the General Directorate of Nature Conservation and National Parks, Ministry of Agriculture and Forestry of the Republic of Türkiye (permit number: 21264211-288.04[Biodiversity Research Permits]-E.1495402, dated 14/05/2019).

## DNA Deposition

The following information was supplied regarding the deposition of DNA sequences:

The DNA sequences are available at GenBank: COI sequences: PV537515—PV537560; ITS2 sequences: PV547651—PV547696.

## Data Availability

Raw data is available in the Supplemental Files.

## Supplemental Information

Supplemental information for this article can be found online at http://dx.doi.org/10.7717/peerj.19870#supplemental-information.

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
