# Peer review of "Evaluating COI and ITS2 dual barcoding for molecular delimitation and taxonomic insights in Arenosetella Wilson, 1932 (Harpacticoida: Ectinosomatidae) along Turkish Coasts"

_PeerJ, doi:10.7717/peerj.19870_

## Round 0.1 · original submission · Minor Revisions

All three reviewers agree that this is a solid piece of work that has the potential to serve as a benchmark for future studies. The study deals with a notoriously difficult family of harpacticoid copepods which is known for its numerous species complexes and for being rife with cryptic and sympatric species. The authors designed an elegant integrative approach to unraveling the relationships between members of a speciose genus of interstitial ectinosomatids combining molecular sequence data and morphology. The manuscript can be accepted for publication subject to minor revision. The authors are encouraged to address the points raised by the reviewers and provide a point-by-point rebuttal when submitting the final version. I suggest the initials C.B. are removed throughout from Wilson's name (including the title) as there is no confusion with his namesake M.S. Wilson. Finally, the authors used MARNA to predict the secondary structure of their ITS2 sequences, however, the MARNA website stated that this software has not been maintained since 2005 and has been superseded by the more advanced tool LocARNA. It needs to be clarified why MARNA and not LocARNA was used and whether any costs were assigned to certain assumptions (e.g. gaps).

Reviewer 1 ·

Basic reporting

The manuscript is well written. I have listed below some ‘soft’ suggestions for improving choice of wording.
16 replace ‘where’ with ‘in which’
22 reword as: Nuclear NDA from a total of 46 individuals were amplified and sequenced for both mitochondrial
31. distribution patterns
35. Replace “confirmed’ with ‘support’
41-42 Be more specific, reword: combining DNA sequences and structure, and morphological….
52 replace ‘where’ with ‘in which’
57 replace ‘high’ with ‘significant’ or ‘considerable’
62 ‘different’ is a vague word and raises the question of how different the populations are. What word for ‘different’ did Mielke (1 975, 1986) use?
72 replace “both sides of the paired bases’ with ‘both strands of the paired bases’
Replace ‘secondary structure are’ with ‘secondary structure that’
73 Coleman, 2009 is listed as Coleman, 2007 in the References section
104 was the 99% ethanol denatured. I guess it was not.
105 replace ‘under’ with ‘using’
126 I think you need an em dash or en dash instead of hyphens
135 insert “and”. … and the aligned dataset….
146 replace casual language ‘scrapped’ with ‘eliminated’
219 insert 2 words: The average GC content…
234-243 Please refer to the particular supplementary file for each statement. For example, on line 240, is the statement about MOTU4a a reference to Supplemental file 3.
317 Please insert a label on Figure 6 as to which seta is seta 4

The literature is well referenced. I would only add:
69 ITS2 is also a population level marker and should be cited as such, with an appropriate reference. Perhaps add some comment, either in the Introduction or Discussion, on the different nature of molecular evolution between the mitochondrial COI gene and the nuclear spacer region and how selection is believe to act on these regions.

Figures and Tables
Overall, the data presented in Figures and Tables is complete and clearly described. Supplementary Figure 4 is especially nice in that the sequence alignment and DNA structure are presented in the same figure, facilitating comparison.
If one believes that a reader should be able to interpret tables and figures without referring to the text, then abbreviations (e.g. MOTU, CBC, ABGD) should be spelled out in the captions.
Figure 2: I do not see any yellow color referred to in the caption.
Figure 5: Image “B” requires more description in the Figure caption.
Figure 6: Label, perhaps using arrows, the particular referred to in the text in lines 316 & 317
I could not find Figure Legends for the Supplementary Figures, nor for the Supplementary Tables, which hampered my understanding of Figures S3 and S4. Axes in Figures S1-3 are missing.
Font size should be increased in Figures S4 so that the particular nucleotides that are referenced can be discerned
Table 3: The reason for use of the Turkish abbreviation ‘ve ark’ is not clear.
100 Table S1 lists 10 localities, not 9

Experimental design

The topic of this paper is exceptionally well suited for publication in Peerj. Harpacticoids are among the most challenging taxa in the Copepoda on which to perform systematic studies due to their morphological stasis and tiny size. This manuscript provides some of the most useful data on harpacticoid systematics. In the absence of mating studies, the molecular and morphological markers used to infer relationships are on target, along with biogeographical data comprises a persuasive case for incipient speciation.
Overall, the descriptions of the methods were presented with exceptional clarity and detail. The only exception was omitting assumptions used to generate secondary structures of ITS2 sequences that would allow the analyses to be repeated. For example, were penalties assigned for gaps?

Validity of the findings

An important strength is the use of both nuclear and mitochondrial markers, as well as ribosomal DNA structure and morphology, locality information Additionally, the morphological descriptions are based on the same specimens used to obtain DNA data. integration of different kinds of data to infer taxonomic and systematic relationships is rare and could serve as a model approach for similar studies.
The sample sizes are sufficient. Standard practices have been used to obtain and interpret the data related to DNA sequences and morphology.

Additional comments

Conclusions:
This is a superbly executed study of molecular and morphological traits of closely related harpacticoids with very well reasoned interpretations of the data. Future studies of copepods could well use this study as an exemplar.
The authors provide very persuasive data for distinct lineages identified as multiple MOTUs. While such data may not be sufficient to define multiple species, the biogeographical separation of the MOTUS adds support to the interpretation of not only interpopulation level variation, but also incipient speciation or several species. Similarly one cannot know for certain that a species complex has been defined within the A. germanica group, as these genetically differentiated lineages might just represent interpopulation level variation and interbreeding studies have not been performed. It would be helpful to elaborate on the findings of Miekle (1975) on line 341 and how they specifically relate to the findings in the present manuscript.
80 It may be premature to refer to Arenosetella populations as ‘a model’ in the sense of a ‘model organism’, because other kinds of data used to infer species relationships, such as interbreeding, are absent. Thus one cannot be definitive, about the degree of reproductive isolation, if any, and thus how the differentiation in the molecular and morphological markers correspond to species relationships as determined by interbreeding studies. This criticism by me is philosophical. Nevertheless, the amassed data present substantial progress toward sorting out population and species relationships. Furthermore, the data are yet another example in a growing body of evidence that genetic differentiation can be maintained in the marine environment, which was once assumed to be comprised of well mixed water masses that inhibit genetic differentiation.

Reviewer 2 ·

Basic reporting

Well-written article with professional English used.

Experimental design

The experiment conducted was aligned with the research question, except for a few sections that need improvement (see attachment).

Validity of the findings

The data generated was well evaluated with robust analysis.

Additional comments

Please make sure the data provided (supplementary) is aligned with the text written in the article (see attachment).

Annotated reviews are not available for download in order to protect the identity of reviewers who chose to remain anonymous.

Reviewer 3 ·

Basic reporting

This study investigates the genus Arenosetella, represented along the Turkish coasts by A. germanica and A. lanceorostrata, using an integrative dual-marker barcoding approach. A total of 46 specimens from the Mediterranean, Aegean, and Black Sea coasts were sequenced for mitochondrial COI and nuclear ITS2 markers.
The manuscript is well-written, clearly structured, and scientifically sound. The experimental design is well-suited to the study objectives, and the combined use of two molecular marker (COI and ITS and CBC analsyes) is highly appropriate for addressing species delimitation in a morphologically conservative taxon such as Arenosetella. The phylogenetic and structural analyses are conducted rigorously, and the results are presented clearly and interpreted with a thoughtful, well-reasoned discussion. The integrative approach, which combines molecular, structural, and morphological details, adds significant value and makes an important contribution to the taxonomy of harpacticoid copepods.

Experimental design

The experimental design is well-suited to the study objectives, and the combined use of two molecular marker (COI and ITS and CBC analsyes) is highly appropriate for addressing species delimitation in a morphologically conservative taxon such as Arenosetella. The phylogenetic and structural analyses are conducted rigorously, and the results are presented clearly and interpreted with a thoughtful, well-reasoned discussion. The integrative approach, which combines molecular, structural, and morphological details, adds significant value and makes an important contribution to the taxonomy of harpacticoid copepods.

Validity of the findings

The integration of sequence-based and structure-based analyses, especially the use of ITS2 secondary structure and compensatory base changes, adds depth and rigor to the species delimitation process. The findings are valid, well-supported by multiple lines of evidence, and reveal significant cryptic diversity within A. germanica, alongside confirming the distinct species status of A. lanceorostrata. This study represents a meaningful contribution to copepod taxonomy and highlights the value of integrative molecular tools in uncovering hidden biodiversity.

---

## Round 0.2 · accepted · Accept

Dear authors,

Thank you for your detailed response to the reviewers' comments. As a result of your careful point-by-point rebuttal, your manuscript no longer needs to be subjected to a second round of reviews and can be accepted in its current format for publication in PeerJ. Congratulations.